# The Potential Connection between Molecular Changes and Biomarkers Related to ALS and the Development and Regeneration of CNS

**DOI:** 10.3390/ijms231911360

**Published:** 2022-09-26

**Authors:** Damjan Glavač, Miranda Mladinić, Jelena Ban, Graciela L. Mazzone, Cynthia Sámano, Ivana Tomljanović, Gregor Jezernik, Metka Ravnik-Glavač

**Affiliations:** 1Department of Molecular Genetics, Institute of Pathology, Faculty of Medicine, University of Ljubljana, 1000 Ljublana, Slovenia; 2Center for Human Genetics & Pharmacogenomics, Faculty of Medicine, University of Maribor, 2000 Maribor, Slovenia; 3Laboratory for Molecular Neurobiology, Department of Biotechnology, University of Rijeka, 51000 Rijeka, Croatia; 4Instituto de Investigaciones en Medicina Traslacional (IIMT), CONICET-Universidad Austral, Buenos Aires B1629AHJ, Argentina; 5Departamento de Ciencias Naturales, Universidad Autónoma Metropolitana Unidad Cuajimalpa, Mexico City 05348, Mexico; 6Institute of Biochemistry and Molecular Genetics, Faculty of Medicine, University of Ljubljana, 1000 Ljubljana, Slovenia

**Keywords:** amyotrophic lateral sclerosis, ALS-related genes, CNS development, neuroregeneration, peripheral biomarkers, non-coding RNAs

## Abstract

Neurodegenerative diseases are one of the greatest medical burdens of the modern age, being mostly incurable and with limited prognostic and diagnostic tools. Amyotrophic lateral sclerosis (ALS) is a fatal, progressive neurodegenerative disease characterized by the loss of motoneurons, with a complex etiology, combining genetic, epigenetic, and environmental causes. The neuroprotective therapeutic approaches are very limited, while the diagnostics rely on clinical examination and the exclusion of other diseases. The recent advancement in the discovery of molecular pathways and gene mutations involved in ALS has deepened the understanding of the disease pathology and opened the possibility for new treatments and diagnostic procedures. Recently, 15 risk loci with distinct genetic architectures and neuron-specific biology were identified as linked to ALS through common and rare variant association analyses. Interestingly, the quantity of related proteins to these genes has been found to change during early postnatal development in mammalian spinal cord tissue (opossum *Monodelphis domestica*) at the particular time when neuroregeneration stops being possible. Here, we discuss the possibility that the ALS-related genes/proteins could be connected to neuroregeneration and development. Moreover, since the regulation of gene expression in developmental checkpoints is frequently regulated by non-coding RNAs, we propose that studying the changes in the composition and quantity of non-coding RNA molecules, both in ALS patients and in the developing central nervous (CNS) system of the opossum at the time when neuroregeneration ceases, could reveal potential biomarkers useful in ALS prognosis and diagnosis.

## 1. Introduction

Amyotrophic lateral sclerosis (ALS), also called Lou Gehrig’s disease, is a complex and genetically heterogeneous neurodegenerative disorder characterized by progressive muscle paralysis reflecting the degeneration of motor neurons (MNs) in the primary motor cortex, corticospinal tract, brainstem, and spinal cord [1]. Particularly, ALS affects lower MNs in the spinal cord and brainstem and upper MNs in the motor cortex [2]. This neuronal degeneration leads to progressive skeletal muscle atrophy and death by respiration failure after 2–5 years from the onset of symptoms [3]. The disease, presenting in middle age, has an incidence of 2 per 100,000 persons per year. The familial ALS forms represent only around 5% of cases, while the majority of cases are sporadic forms, which are mostly phenotypically indistinguishable from familial forms, suggesting the involvement of common pathophysiological pathways. Numerous studies have been focused on revealing the variety of neurodegenerative processes underlying ALS, including oxidative stress, mitochondrial impairment, excitotoxicity, alteration in the Ca^2+^ homeostasis, mitochondrial dysfunction, growth factor deficiency, defective axonal transport, disrupted proteostasis, and RNA metabolism [4,5,6,7,8], involving both neuronal and non-neuronal cells [9]. Indeed, ALS is a multifactorial disease with diverse etiology and pathogenesis, and suspected immune-mediated mechanisms remain, however, controversial [10].

Currently, there is no definitive therapy, and the drugs approved for ALS treatment, Riluzole and Edaravone, provide only modest benefits and only in some patients [11,12]. Recently, promising new therapeutic approaches for the treatment of ALS have emerged, including gene therapy and cell therapy. ALS-directed gene therapeutic strategies include antisense oligonucleotides, RNA interference, CRISPR, adeno-associated virus (AAV)-mediated trophic support, and antibody-based methods, some of which have reached human clinical trials [13]. To develop safe therapeutic strategies, it is of major importance to consider the normal functions of the manipulated genes and the potential contribution of gene loss-of-function to ALS [14]. In restorative approaches using transplanted stem cells or neural progenitor cells, the major problem considers the complexity of the functional networks required to produce walking, which is produced via a complex developmental program that is inactivated in adulthood [15].

Both the diagnosis and treatment of the disease are currently insufficient, primarily due to deficient knowledge of genetic and molecular causes of the disease as well as pathological pathways involved in the disease progression. Here, we propose a possible connection of ALS with neuroregeneration and aberrant early central nervous system (CNS) development [16,17], where the complex pathophysiology could be orchestrated by several non-coding RNAs that also could present potentially useful biomarkers in ALS prognosis and diagnosis. Moreover, the miRNA and lncRNA molecules known to control the expression of the ALS-related genes of interest have been co-located both in the peripheral blood of ALS patients, as well as in brain and spinal cord tissues [18,19], giving a strong indication of possible involvement and importance in the control of both CNS development and ALS pathophysiology.

## 2. Genetics of ALS

Although there is not a final criterion for familial ALS, the general agreement is accepted that the presence of the disease in first-or second-degree relatives represents a family disease [20]. Many familial types of ALS have a classic pattern of Mendelian inheritance, mainly in an autosomal-dominant manner with a high degree of penetration. However, cases of recessive and X-linked recessive inheritance are also described [21]. In 1989, the first gene locus on chromosome 21 (*ALS1*), which is associated with a dominant type of familial ALS, was discovered [22]. On this locus, the *SOD1* gene and numerous mutations in it in connection with ALS were later identified [23]. The ALS disease is related to many different genes and already more than 150 genes are currently described in the accessible ALSoD database (http://alsod.iop.kcl.ac.uk/) (accessed on 1 July 2022). However, the vast majority of mutations in these genes were discovered in connection with the disease in individual cases; therefore, those are currently classified only as candidate genes. On the other hand, the changes in only four genes, *SOD1* [23], *FUS* [24,25], *TARDBP* [7,26,27], and *C9orf72* [28,29], represent a high risk for the development of ALS, which has been confirmed by numerous studies. The FUS and TDP-43 mutations were connected with nuclear transport and the metabolism of RNA. Mutations in genes *SOD1*, *TARDBP* (TDP-43), and *FUS* together occur in 20–30% of the familial type of the disease [30], and mutations in the *SOD1* gene alone affect up to 20% of patients with the autosomal dominant type of familial ALS and 2% of patients with the sporadic type of ALS [31]. In hereditary types, the mutation, which causes the disease, is detected in approximately two-thirds of patients and only in about 11% of patients with the sporadic type [32,33]. Sporadic types may be caused by mutations with low penetration, de novo mutations, different genetic variations, and epigenetic events [34]. Clinically, sporadic and familial types do not distinguish between each other, but in comparison with sporadic, the familial type of the disease occurs at an earlier age of life [35], approximately a decade before the sporadic type [31], lasts longer, and women and men get ill with the same frequency [35,36]. Up to 50% of all patients with ALS also express symptoms of frontotemporal dementia (FTD), which leads to the degradation of neurons in frontotemporal brain lobes [37]. Interestingly, it was discovered that some of the ALS-specific genes, *UBQLN2* [21] and *C9orf72* [28,29], are responsible for the combination of ALS-FTD [38]. The discovery of the involvement of *C9orf72* in the year 2011 represented a new milestone in the research because it is one of the most common genetic causes of the disease’s development. The expansion of hexanucleotide (GGGGCC) repeats in the first intron of *C9orf72* is not only one of the most common mutations in connection with ALS but also the second most common in FTD cases [39]. In addition, this discovery strengthened the hypothesis that the pathogenesis of ALS is carried out by several different genetic and molecular pathways [4]. Some of the less-frequently mutated genes associated with ALS, of which some are also linked to other neurodegenerative diseases, include *OPTN* [40], *VCP* [28,41,42], *SQSTM1* [43], *FIG4* [44], *ATXN2* [45], *DAO* [46], *SPG11* [47], *PFN1* [48], *VAPB* [49], *ALS2* [50], *SETX* [51], and *ANG* [52]. Recently, a link between the haploinsufficiency of *TBK1* and ALS has also been discovered [53], and a kinesin family member gene (*KIF5A*) was recognized as a novel gene associated with ALS [54,55] as well as a new ALS risk gene *C21orf2* [56] and 15 risk loci with different genetic architecture and neuron-specific biology [16]. Moreover, in patients with ALS, some studies have shown the changing patterns of methylation in ALS-related genes [57,58,59] and connected the engagement of the epigenetic mechanisms in motor neuron apoptotic cell death through DNA methylation, which is essential during the development of the nervous system and could be relevant to human ALS pathobiology [60]. The results of several studies suggest that epigenetic changes (through methylation and noncoding RNAs) may represent new biomarkers and contribute to the development of new ALS therapies (reviewed in [61,62,63]). Moreover, we propose that the overlap and correlation in the expression of the selected biomarkers during early CNS development and in ALS patients could narrow down the number of potential candidates, as well as open new insights into disease pathogenesis.

## 3. MiRNA and lncRNA Controlling ALS

The expression of genes depends on several different processes including variations in coding DNA sequences and the expression of non-coding RNAs (miRNAs, lncRNAs).

MiRNAs are a class of small, single-stranded, non-coding RNAs that play important roles in gene-regulation by targeting mRNAs for cleavage or translational repression [64]. Since each miRNA can regulate up to hundreds of mRNA targets, it is possible that the transcriptome alterations detected in ALS patients derive, at least partially, from the disruption of miRNA networks. Indeed, studies using miRNA microarrays or RNA deep sequencing have revealed extensive changes in the expression of more than a hundred miRNAs in ALS [18,19,65,66,67,68]; however, the detection of the precise roles of specific miRNAs and/or combinations of miRNAs for the development of disease-specific biomarkers is a focus of current research [69,70]. In current review studies, more commonly detected deregulated miRNAs associated with ALS have been collected and include let-7b, miR-9, miR-16-5p, miR-124a, miR-128, miR-132, miR-133b, miR-143, miR-451, miR-181, miR-183, miR-206, miR-338-3p, and miR-638 [18,19,71,72,73]. However, data on miRNAs as biomarkers in ALS are sometimes contradictory and currently still not clear enough for translation into clinical practice.

LncRNAs are RNA transcripts greater than 200 nucleotides that lack an open reading frame and have little coding potential. Although the functions of the vast majority of lncRNA transcripts remain unknown, it is assumed that they are possible regulators of biogenesis, cellular cycles, and differentiation, and are involved in nervous system development and neurological diseases [74]. LncRNAs can act both as the epigenetic regulators of target genes and as components of an extensive, unexplored network of interacting RNAs involving miRNAs and mRNAs.

The roles of lncRNA in ALS have just started to be explored and a small number of lncRNAs associated with ALS demonstrate that there is still much to do to identify and understand their role in ALS [75]. The RNA-seq analyses of sALS and fALS patients revealed the presence of lncRNAs differentially expressed both in blood mononuclear cells and in the spinal cord, including *NEAT1,* *MALAT1*, and *MEG3* [76,77,78]. In the study of Yu et al., lnc-DYRK2-7: 1 and lnc-POTEM-4: 7 were specifically down-regulated in affected subjects compared to healthy controls [79].

## 4. Novel ALS GWAS Risk Loci and Their Connection with CNS Development and Neuroregeneration

It is very likely that there are multiple genetic alterations on different genomic regions that carry information for the coding and non-coding RNAs necessary for the emergence of complex diseases. Thus, a complex cross-ancestry genome-wide association study (GWAS) was recently performed, combined with whole-genome sequencing and a large cortex-derived expression quantitative trait locus dataset. The data revealed 15 ALS-associated risk loci and the related genes, indicating perturbations in vesicle-mediated transport and autophagy and cell-autonomous disease initiation in glutamatergic neurons [16] (Table 1).

Neuroregeneration is a complex process, influenced by numerous cellular and molecular mechanisms, yet not fully understood. Understanding why neuroregeneration fails in the adult mammalian CNS would give the possibility to alter the key pathways to promote cell survival and axon regeneration and possibly alter the outcome of neurodegenerative diseases, including ALS [80]. The comprehensive analysis of the proteins that change during the early developmental period when neuroregeneration in CNS stops being possible revealed the abundance of the proteins related to neurodegenerative diseases, including the proteins coded by the genes related to the ALS risk loci [17]. Those results connect ALS with neurodevelopment and neuroregeneration, with the possible identification of new diagnostic and therapeutic targets.

## 5. Studying Neuroregeneration in Opossum *Monodelphis domestica*

One of the major challenges of modern neurobiology concerns the inability of the adult mammalian CNS to regenerate and repair itself after injury or after neuronal loss in neurodegenerative diseases, although new therapies supporting compensation mechanisms could give alternative solutions. It is still unclear why the ability to regenerate the CNS is lost during evolution and development and why it becomes very limited in adult mammals. A convenient model to study the cellular and molecular basis of this loss is the neonatal opossum (*Monodelphis domestica*). Opossums are marsupials that are born very immature with the unique possibility to successfully regenerate the postnatal spinal cord after injury in the first two weeks of their life, after which this ability abruptly stops at 14 days in cervical spinal segments and at 17 days in less mature lumbar spinal segments [81]. Thus, neonatal opossums represent a unique opportunity to achieve and study mammalian CNS that can regenerate, without the need for the invasive intrauterine surgery of pregnant females (as is necessary for other mammalian laboratory animals, such as mice or rats). In addition, the tiny neonatal opossum CNS can be maintained in culture in its entirety and this preparation is similar to the intact animal in its ability to regenerate [82]. Moreover, in isolated preparations, reflex activity and neurogenesis continue, and the structure remains normal in appearance, with minimal cell death [83].

Even though our understanding of molecular and cellular mechanisms that promote or inhibit neuronal regeneration is expanding [84], it is still unclear what are the key differences between the neuronal systems that can and cannot regenerate and how they can be manipulated to revert the outcome. For reasons yet unknown, with development and age, the mammalian CNS loses its capacity for functionally relevant repair after injury, as is observed on the evolutionary scale [81]. Although the CNS is more plastic early after birth than in the adult, the postnatal mammalian CNS usually displays limited regenerative capacities, and the precise cellular and molecular basis for this developmental loss are mostly unclear. In the short-tailed gray opossum, *Monodelphis domestica*, newly born animals correspond roughly to 12-day mouse or rat embryos [85], and they have the ability to fully functionally regenerate the cervical spinal cord until the postnatal day 12, while in the less-mature lumbar segments, regeneration continues until postnatal day 17, approximately [86]. Thus, opossums represent the unique opportunity to achieve and study mammalian CNS that can regenerate without the need for intrauterine surgery on pregnant mothers. In previous studies [87,88], differentially expressed genes were identified in opossum spinal cords during the critical period of development when regeneration stops being possible, revealing the molecules involved in nucleic acid management, protein synthesis and processing, the control of cell growth, structure and motility, cell signaling, extracellular matrix molecules, and their receptors. The transcriptomic data were upgraded with proteomic research [17], to select the overlapping molecules as the most promising candidates controlling regeneration, to be tested in functional studies.

## 6. ALS-Related Proteins Change in Opossum Spinal Cord When Regeneration Ceases

Among the proteins that are differentially distributed in opossum spinal tissue that can and cannot regenerate after injury, we identified the numerous proteins related to neurodegenerative diseases, such as Huntington’s, Parkinson’s, and Alzheimer’s disease [17]. Interestingly, the ALS-related proteins were detected, among which were those connected to the 15 risk loci linked to ALS [16] (Table 2). Several proteins are present uniquely in the regenerating spinal cord tissue taken from 5-day-old opossums (P5), while the others are conversely present only in the non-regenerating opossum tissue taken from the older animals (P18). Moreover, we have found different ALS-related proteins to be present in both regenerating (P5) and non-regenerating (P18) spinal cords, but to a dissimilar degree. Their functional classification is shown in Figure 1.

To our knowledge, this is the first time that the ALS-related genes or proteins have been found to change their expression during the early postnatal development in mammalian spinal cord tissue during the checkpoint when neuroregeneration ceases to be possible. These results indicate that the proteins related to ALS, and in general to neurodegeneration in aged CNS, could have an important physiological role during postnatal neurodevelopment, particularly that they could be involved in the molecular and cellular mechanisms underlying neuroregeneration [89].

We have found some of the most well-characterized ALS high-risk factors, such as SOD1 and OPTN proteins, to be upregulated in non-regenerating (P18) opossum spinal cords, unlike FUS, which was upregulated in regenerating (P5) opossum spinal cords (Table 2). TBK-1 was found to be uniquely expressed in P18 spinal cords. The time course of their expression during postnatal CNS development in marsupials such as *M. domestica* remains largely unexplored. Further, UNC13A (or Munc13-1 in mammals), another ALS-related protein [16,90,91], was expressed exclusively in the P18 opossum spinal cord (Table 2). UNC13A is essential for synaptic vesicle maturation during exocytosis as a target of the diacylglycerol second messenger pathway and for the neurotransmitter release of most glutamatergic but not inhibitory GABA-mediated synapses [92,93]. Its unique expression in the P18 spinal cord correlates with the developmental stage at which the neurogenesis is almost completed [94] and newly generated neurons have to establish and consolidate their synaptic connections with other neurons.

## 7. KIF5A and Kinesin Family Member Proteins Expressed in Developing Opossum Spinal Cord

Neurons are highly polarized cells with neurites that can extend over long distances from soma. Axonal transport is fundamental during neuronal development as well as in mature neurons, allowing efficient intra- and intercellular communication [95,96]. Kinesins are microtubule-based motor proteins that play a central role in the axonal transport of several cargos including organelles such as mitochondria, neurofilaments, receptors for neurotransmitters including glutamate [97], and granules containing both RNA and RNA-binding proteins [55].

KIF5A, KIF5B, and KIF5C are mammalian heavy chain isoforms that belong to the Kinesin-1 family and all are expressed in neurons [95,98]. *KIF5A* was recently identified as a novel ALS gene [16,54,55]. Missense mutations of *KIF5A* cause the defective anterograde transport of cargo along dendrites and axons. Deficiency in KIF5A expression and cargo binding has been associated with the accumulation of phosphorylated neurofilaments and amyloid precursor proteins within neuronal cell bodies, and subsequent neurodegeneration, in patients with multiple sclerosis [55].

The proteome analysis of the developing opossum spinal cord revealed 12 kinesins or kinesin-like proteins (Table 2). For instance, KIF5C was found to be up-regulated in P5 spinal cords, while KIF5B was up-regulated in P18 spinal cords. These expression profiles are in agreement with KIF5 expression in 2-week-old mice since both KIF5B and KIF5C were upregulated in axon-elongating neurons, with KIF5C highly enriched in lower motor neurons [98]. KIF5B is expressed in glial cells [98], and this further confirms the correlation between the onset of gliogenesis in P18 opossums [94] and its upregulation.

In addition to Kinesin-1 family members, KIF1B, KLC2, KLC4, and KIF21B were up-regulated in P5 spinal cords, while KIF1A, KIF2A, KIF3A, KIF3B, and KIFAP3 were up-regulated in the P18 spinal cord. KIF15 was found to be unique in the P5 opossum spinal cord. This observation is in agreement with its co-localization with microtubules in dendrites and in growth cones, in particular in migratory neurons [99], because at that developmental stage (P5), extensive neurogenesis occurs in the opossum CNS [94,100].

## 8. NEK1 and CFAP410 and Interacting/Related Partners

*NEK1* and *CFAP410* were identified as ALS risk factors/loci in humans (Table 1) [16,101,102]. The protein encoded by *NEK1* is involved in cell-cycle regulation [103], essential during the transition from the proliferative (undifferentiated) state of neuronal progenitors to the post-mitotic (differentiated) state of neurons. The proteomic analysis of the opossum P5 and P18 spinal cord proteome did not identify proteins encoded by the *NEK1* gene. However, NIMA-related kinase 9 (Nek9), a member of the same family of serine/threonine protein kinases involved in mitosis [104], was identified as unique to the P18 opossum spinal cord.

*MAP2K1*, encoding for mitogen-activated protein kinase 1, is an important paralog of *NEK1*. MAP2K1 was identified in both P5 and P18 opossums, with higher expression in P18 (1.08 times more). We have recently shown [105] that both MAPK/p38 and JNK/c-Jun signaling pathways are involved during neurite outgrowth and neuronal network formation as well as during regeneration after injury through the interaction and induction of activating transcription factor 3 (ATF3) [106,107].

Kinesin-1 adapter fasciculation and elongation protein zeta 1 (FEZ1), known to be found in the centrosomal complex with NEK1 and involved in axonal development [108], was found to be expressed only in P5 opossums (LFQ intensity 5.6 × 10^6^).

In addition to the cell cycle, NEK1 is also involved in cilia formation and maintenance as well as in the regulation of cell morphology and cytoskeletal organization [103,104] and is in direct interaction with CFAP410, another ALS risk factor [16,109]. Proteomic analysis in opossums did not reveal it, but cilia- and flagella-associated protein 20 (CFAP20), with a mainly unknown function, was found to be unique to P5 (Table 2). Moreover, NEK1 is a direct interacting partner with KIF3A (upregulated at P18), in which dysfunction induces developmental defects in neuronal migration and differentiation, delays in neural stem cell-cycle progression, and failures in interkinetic nuclear migration [110].

These data strongly indicate that common pathways are involved in both development, regeneration, and degeneration. Being unique to P5 opossums (equivalent to E16 mice or E18.5 rat embryos [85]), FEZ1 and CFAP20 expression confirms their involvement in early CNS, with ongoing neurogenesis. It is striking that the time when neuroregeneration ceases in the opossum spinal cord [82,86] overlaps with the switch from neurogenesis to gliogenesis, once the formation of the six-layered cortex is accomplished, between P18 and P20 [94]. This fact could be in connection with the discovery that ALS is a cell-autonomous disease with initiation in glutamatergic neurons [16]. For example, the UNC13A, uniquely expressed in the P18 opossum spinal cord, could be involved in or responsible for perturbation in glutamatergic neurotransmission, in addition to dysfunctions in kinesin-mediated transport [111,112].

## 9. Could Neurodegeneration Be Considered a Failure of Neuroregeneration?

Common ALS-related genes/proteins identified in [16] (Table 1) and in the proteome analysis of developing opossum spinal cord [17] (Table 2) are summarized in Figure 2. It is still to be tested if the ALS-related genes/proteins have an impact on neuronal regeneration. Particularly, it is to be studied if the neurodegeneration could be considered as a failure of neuroregeneration [113] and possibly as the regress of injured neurons to an embryonic or early postnatal transcriptional growth state [114]. The supporting evidence for this point of view has been found from recent investigations, where induced pluripotent stem cells (iPSCs) or induced neurons (iNs) that were derived from AD patients showed a de-differentiated phenotype, reminiscent of an immature (i.e., progenitor-like) neuronal state. These cells also show signs of a cell cycle re-entry [115,116]. These observations suggest that the process of de-differentiation in response to injury represents the link between neuronal degeneration, development, and regeneration and this might apply to ALS as well.

## 10. Do the Same Non-Coding RNAs Control Neuroregeneration and ALS-Related Neurodegeneration?

The non-coding RNAs are known to have an important role in controlling CNS axon regeneration [117] and neurogenesis [118], but also neurodegeneration [119,120]. Thus, it would be of great interest to understand if the non-coding RNAs that control the expression of ALS-related genes might have a role in neuroregeneration. Particularly, it would be interesting to reveal if they change their expression during the developmental period when neuroregeneration in the mammalian spinal cord stops being possible. The construction of an mRNA–miRNA–lncRNA network that would contain the overlapping molecules that have a role both in ALS pathobiology and also in early CNS development and in neuroregeneration and neurodegeneration, would open the scenario for narrowing down the potential candidates for ALS-detecting biomarkers and also give new insight into the molecular basis of ALS, with the possibility of the early diagnosis of the disease in childhood.

In Appendix A, we have listed the ALS marsupial-related genes from Table 2 and up to five of their most-predicted interacting human miRNA and lncRNA molecules, indicating their expression in the blood, muscles, and spinal cord. In addition, in Table 3, we specifically collected, from Appendix A, those human miRNAs that potentially control the expression of genes that code proteins differentially distributed in P5 and P18 opossum spinal cords and which have already been reported to be dysregulated in ALS patients.

It is striking that not only the genes coding the ALS-related proteins are both present in the peripheral blood of ALS patients and in the opossum spinal cord, but also their interacting miRNA and lncRNA molecules, making them strong candidates for ALS biomarkers (Appendix A, Table 3, and Figure 3). Several of those molecules have already been recently connected to neuroregeneration. For example, the downregulation of the hsa-miR-124-3p, which is related to KIF1B and PTPRZ1 (Table 3 and Figure 3), is known to promote the neural stem activation through the activation of Wnt/β-catenin signaling in spinal cord neural progenitor cells [121], having beneficial effects on neuroregeneration [122]. Next, the hsa-miR-200c-3p, related to PTPRD and TBK1 (Table 3 and Figure 3), was found to be a key factor that promotes successful spinal cord regeneration in axolotls, regulating the stem cell identity [123]. The hsa-miR-21-5p, related to KIFAP3, was recently shown to be involved in the promotion of structural and functional recovery in sciatic nerve injury [124]. The hsa-miR-222-3p, related to COG2, KIF5C, and KIF3B (Appendix A, Table 3, and Figure 3) enhanced the neuronal differentiation of neural stem cells and thus the combination of nanofibrous scaffolds with miR-222 is proposed as a promising approach for neural tissue regeneration [125]. The hsa-miR-615-3p, related to COG4, FUS, and RPSA (Table 3 and Figure 3), was shown to negatively regulate immunoglobulin-like domain-containing NOGO receptor-interacting protein 1 (LINGO-1), having a beneficial effect on motoneuron loss, nerve regeneration and myelination, reduced astrocyte activation, the regulated differentiation of neuronal stem cells, and others, facilitating function recovery after peripheral nerve injury and spinal cord injury [126,127]. The hsa-miR-20a-5p, related to GAK (Appendix A) was shown to promote axonal regeneration in dorsal root ganglion neurons [128]. The hsa-miR-21-5p, related to KIFAP3 (Table 3), was shown to promote Schwann cell proliferation and axon regeneration during injured nerve repair [129].

For the other non-coding RNA molecules listed in Appendix A, the role in neuroregeneration and CNS development is still to be studied, determining their usefulness as eventual ALS biomarkers.

The long non-coding RNA molecules are very little known in marsupials, especially in *Monodelphis domestica*. Even though the lncRNAs are widely expressed, their low conservation at the sequence level makes the revelation of their evolutionary history often challenging [130]. However, marsupials contain the paraspeckles, nuclear bodies that in human and mouse cells are assembled around an architectural NEAT1/MENe/b lncRNA, which are tissue-specific, stress-responding nuclear bodies, illustrating their structural and functional continuity over 200 million years of evolution throughout the mammalian lineage [130].

RNA network visualization was performed using the software package CytoScape 3.8.2. [131] using default CytoScape settings and freely available style options.

## 11. Future Perspectives and Conclusions

There is an urgent need for biomarkers useful for the diagnosis and classification of ALS disease. Non-coding RNA molecules are emerging as the key players in CNS development as well as in the pathogenesis of numerous neurodegenerative diseases. Studying the changes in their spatial and temporal expression in different neuronal and glial cell types, during normal brain development and aging, as well as their aberrant expression in pathological states, will help to pinpoint the molecules that have the potential as biomarkers useful in the diagnosis and prognosis of different disorders, including ALS. The differential expression of the gene, as well as the non-coding RNAs in the blood of ALS patients, and the comparison with their expression during the early mammalian CNS development, could give hints towards the resolution of the origin of the disease.

## Figures and Tables

**Figure 1 ijms-23-11360-f001:**
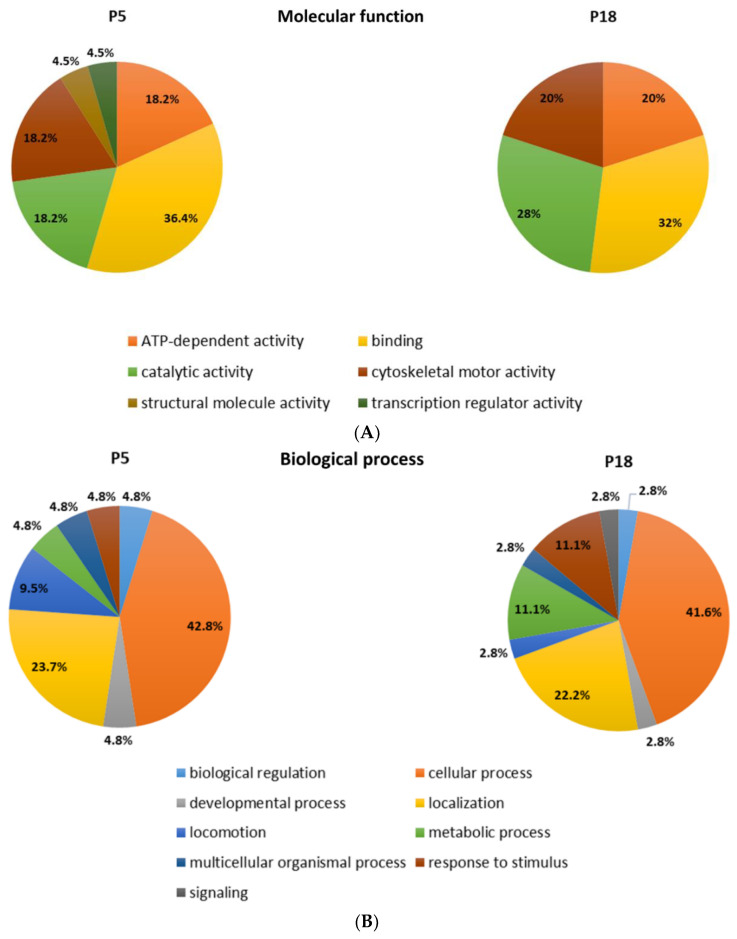
Functional classification of the proteins identified by MS as differentially distributed in the opossum P5 and P18 spinal cords. Proteins were classified based on assumed molecular function (**A**), biological process (**B**), or protein class (**C**).

**Figure 2 ijms-23-11360-f002:**
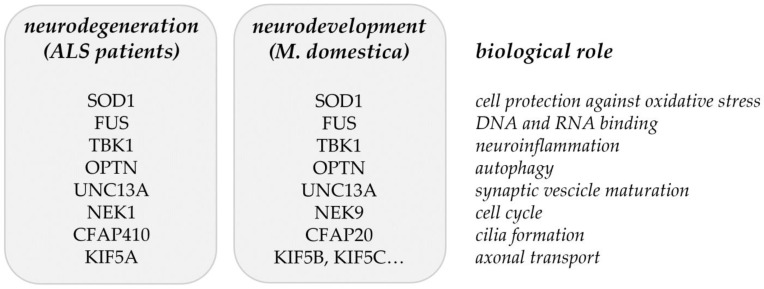
Matching genes/proteins identified through the common and rare variant association analyses from ALS patients [16] and the proteomic analysis of developing opossum spinal cord [17].

**Figure 3 ijms-23-11360-f003:**
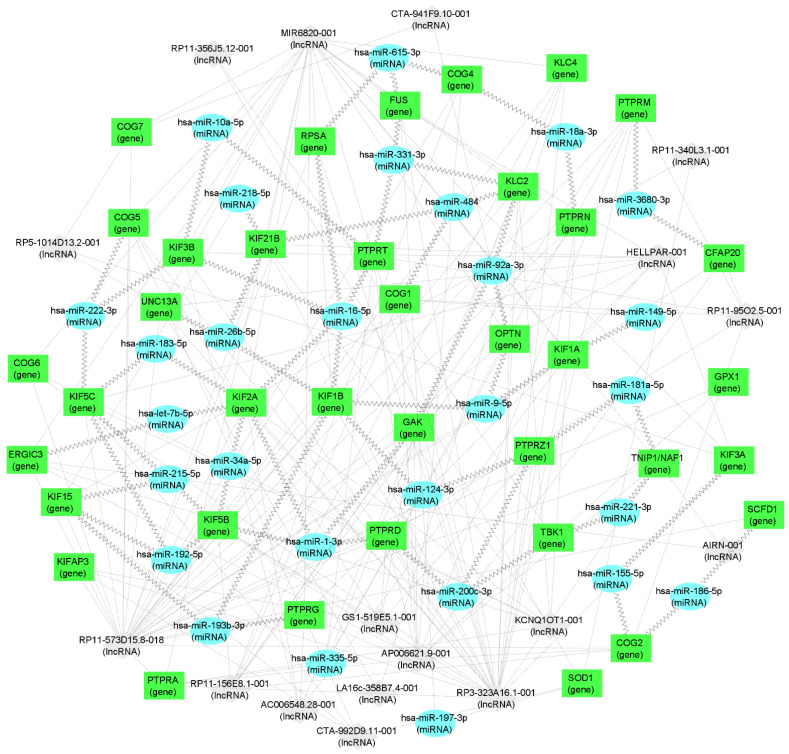
**mRNA–miRNA–lncRNA network visualization.** Green rectangles represent genes (i.e., mRNA), blue ellipses represent miRNA, and gray diamonds represent lncRNA. Jagged connections between RNA nodes represent interactions confirmed in whole blood while dotted lines represent interactions in different tissues. RNA nodes with fewer than two connections were not retained for visualization.

**Table 1 ijms-23-11360-t001:** List of the genes related to the 15 risk loci in ALS identified through the common and rare variant association analyses [16].

Gene	ID (GenBank)
*SOD1*	6647
*C9orf72*	203228
*NEK1*	4750
*PTPRN2*	5799
*FUS*	2521
*COG3*	83548
*ERGIC1*	57222
*TARDBP*	23435
*TBK1*	29110
*OPTN*	10133
*SLC9A8*	23315
*SPATA2*	9825
*GPX3*	2878
*TNIP1*	10318
*CFAP410*	755
*KIF5A*	3798
*MOBP*	4336
*RPSA*	3921
*SCFD1*	23256
*UNC13A*	23025

**Table 2 ijms-23-11360-t002:** Proteins related to ALS and detected by mass spectrometry as differentially distributed in P5 and P18 opossum spinal cords.

**Proteins unique for P5 spinal cords**	**ID**	**Gene symbol**	**LFQ** **Intensity**
Nuclear assembly factor 1 ribonucleoprotein	K7E2M8	*TNIP1/NAF1*	1.24 × 10^7^
Kinesin family member 15	F6S782	*KIF15*	8.6 × 10^6^
Cilia- and flagella-associated protein	F6R525	*CFAP20*	7.45 × 10^6^
Component of oligomeric Golgi complex 4	F6T7R4	*COG4*	1.64 × 10^7^
**Proteins unique for P18 spinal cords**			
Tau tubulin kinase 1	F7FPA7	*TBK1*	7.2 × 10^6^
Unc-13 homolog A	F6W9P5	*UNC13A*	1.85 × 10^7^
Component of oligomeric Golgi complex 1	F7F869	*COG1*	8.05 × 10^6^
Component of oligomeric Golgi complex 2	F6ZMV7	*COG2*	5.2 × 10^7^
Component of oligomeric Golgi complex 5	F7ERT4	*COG5*	6.9 × 10^6^
Component of oligomeric Golgi complex 7	F7BTD1	*COG7*	1.4 × 10^7^
Protein tyrosine phosphatase, receptor type T	F7GH48	*PTPRT*	1.71 × 10^7^
Protein tyrosine phosphatase, receptor type N	F6RCL7	*PTPRN*	6.7 × 10^7^
Protein tyrosine phosphatase, receptor type M	F6ZQ35	*PTPRM*	4.25 × 10^6^
**Proteins up-regulated in P5 spinal cords**	**ID**	**Gene symbol**	**Fold change**
Sec1 family domain containing 1	F7BSJ8	*SCFD1*	1.14
40S ribosomal protein SA	F7BC17	*RPSA*	1.43
ERGIC and Golgi 3	F6WZ15	*ERGIC3*	3.63
FUS RNA binding protein	F6WGL5	*FUS*	1.7
Kinesin family member 1B	F7EJI5	*KIF1B*	1.31
Kinesin-like protein	F7GBT8	*KIF5C*	1.02
Kinesin light chain 2	F7B8A8	*KLC2*	1.07
Kinesin light chain 4	F7FP03	*KLC4*	1.42
Kinesin family member 21B	F7A9R7	*KIF21B*	1.11
**Proteins up-regulated in P18 spinal cord**			
Superoxide dismutase (Cu-Zn)	F6VK78	*SOD1*	1.53
Cyclin G-associated kinase	F7CA71	*GAK*	1.23
Optineurin	F6R1Z3	*OPTN*	1.49
Kinesin family member 1A	F6PG86	*KIF1A*	1.04
Kinesin-like protein	F6Y7G9	*KIF2A*	1.28
Kinesin-like protein	F6SD95	*KIF3A*	1.27
Kinesin-like protein	F6RWN1	*KIF3B*	1.35
Kinesin-like protein	F7BJ22	*KIF5B*	1.05
Kinesin-associated protein 3	F7GBK2	*KIFAP3*	1.33
Glutathione peroxidase	F7CS77	*GPX1*	1.01
Component of oligomeric Golgi complex 6	F6SS12	*COG6*	1.02
Protein tyrosine phosphatase, receptor type D	F6S1W5	*PTPRD*	1.2
Protein tyrosine phosphatase, receptor type G	F6Z7H9	*PTPRG*	1.77
Protein tyrosine phosphatase, receptor type Z1	F6ZVL3	*PTPRZ1*	2.8
Receptor-type tyrosine-protein phosphatase	F7G6B6	*PTPRA*	1.73

**Table 3 ijms-23-11360-t003:** miRNAs that regulate the expression of genes that code proteins differentially distributed in P5 and P18 opossum spinal cords and detected in association with ALS. (data from miRTarBase (https://miRTarBase.cuhk.edu.cn/) (accessed on 1 July 2022) and [18,19,71,72,73]).

miRNA	Genes
hsa-miR-1-3p	*GAK, KIF2A, KIF5B, PTPRD*
Let-7b-5p	*ERGIC3, KIF2A*
hsa-miR-10a-5p	*PTPRT, KIF3B*
hsa-miR-9-5p	*OPTN, KIF1A, KIF1B*
hsa-miR-16-5p	*PTPRT, RPSA, KIF1B, KIF2A, KIF3B*
hsa-miR-18a-3p	*PTPRN, COG4*
hsa-miR-21-5p	*KIFAP3*
hsa-miR-26-5p	*KIF1B, KIF21B, UNC13A*
hsa-miR-34a-5p	*KIF2A, KIF5B*
hsa-miR-92a-3p	*KLC2, GAK, OPTN*
hsa-miR-124-3p	*KIF1B, PTPRZ1*
hsa-miR-149-5p	*COG1, KIF1A*
hsa-miR-155-5p	*KIF3A, COG2*
hsa-miR-181a-5p	*TNIP1/NAF, PTPRZ1*
hsa-miR-183-3p	*KIF5C, KIF2A*
hsa-miR-186-5p	*COG2, SCFD1*
hsa-miR-192-5p	*KIF5B, KIF15*
hsa-miR-193b-3p	*KIF15, KIF1B, PTPPG*
hsa-miR-197-3p	*SOD1, FUS*
hsa-miR-200c-3p	*PTPRD, PTPRZ1, TBK1*
hsa-miR-206	*SOD1*
hsa-miR-218-5p	*KIF15, KIF21B*
hsa-miR-221-3p	*TNIP1/NAF, TBK1*
hsa-miR-331-3p	*FUS, KLC2, PTPRT*
hsa-miR-335-5p	*PTPRT, PTPRM, KLC4, OPTN, PTPRA*
hsa-miR-615-3p	*COG4, RPSA, FUS*

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
