# Peer review of "The Potential Connection between Molecular Changes and Biomarkers Related to ALS and the Development and Regeneration of CNS"

_ijms, 2022, doi:10.3390/ijms231911360_

Round 1
Reviewer 1 Report
Molecular basis of ALS and potential biomarkers – 2 the insight from the developmental studies
very interesting article, however, requiring rearrangement
Title: the title does not fully reflect the issues covered in the test
Introduction
line 54-55 „Currently there is no resolutive therapy and the drugs approved for ALS treatment, Riluzole and Edara- 54 vone, provide only modest benefits and only in some patients [11,12].”- recently, new therapies for the treatment of ALS have emerged, such as the use of cell therapy. This topic should be described more
line-60-63 „ Namely, proteins coded 60 by the genes that have recently been associated with ALS through GWAS studies [13] have been also de- 61 tected among the proteins that change their levels during the early postnatal development when neuro- 62 regeneration in mammalian spinal cord ceases [14]. This passage should be removed from the introduction as it is discussed extensively in the next chapter
miRNA, lncRNA and circRNA controlling ALS -too extensive
Novel ALS GWAS risk loci and their connection with CNS development and neuroregeneration
line200-204 The comprehensive analysis of the proteins that change during the early developmental period when neu- 200 roregeneration in CNS stops being possible, reveiled the abundance of the proteins related to neurodegen- 201 erative diseases, including the proteins coded by the genes related to the ALS-risk loci [14] -do I know what this process is caused by?
Studing neuroregeneration in opossum Monodelphis domestica
line 207-208 One of the major challenges of modern neurobiology concerns the inability of the adult mammalian CNS to 207 regenerate and repair itself after injury or after neuronal loss in neurodegenerative diseases - is it so sure it is? or are we looking for new therapies supporting compensation?
NEK1 and CFAP410 and interacting/related partners -this chapter should be shortened because some information is repeated
Figure 2 unreadable
Reviewer 2 Report
The review article titled "Molecular basis of ALS and potential biomarkers - the inisght from the developmental studies" describes and analyses results regarding key molecules that may serve as ALS biomarkers or are mechanistically involved in ALS pathogenesis.
The article discusses interesting findings from opossum studies that link neuroregeneration, neurodevelopment, and neurodgeneration.
The asrticle seems to mix aspects of a review article while presenting/analyzing original data.
The opssum model data is certainly highly interesting in regard to nueodevelopment and neuroregeneration. Yet the model and its use need to be introduced in a borader since for a general audience that might not be familar with it.
The text requires careful editing (punctutation, grammar, style).
It is not quite clear why the authors chose the list of 15 ALS-associated gene/proteins. There are certainly more confirmed genes/proteins closely associated with ALS.
Additional tables and figures (diagrams) will help to communicate major points more effectively
Round 2
Reviewer 2 Report
The revised version of the manuscript is much improved and contains some of the information required to appreciate the opossum model.
Many language problems have also been resolved.